# The Diet as a Modulator of Tumor Microenvironment in Colorectal Cancer Patients

**DOI:** 10.3390/ijms24087317

**Published:** 2023-04-15

**Authors:** Manuel Collado, Marién Castillo, Gemma Julia Muñoz de Mier, Carolina de la Pinta, Cristina Peña

**Affiliations:** 1Medical Oncology Department, Ramón y Cajal University Hospital-IRYCIS, Alcalá University, 28034 Madrid, Spain; manualmansa98@gmail.com; 2Facultad de Ciencias de la Salud, Universidad Alfonso X El Sabio (UAX), Avenida de la Universidad, 1, 28691 Villanueva de la Cañada, Spain; maricasa@uax.es (M.C.); gmunodem@uax.es (G.J.M.d.M.); 3Radiation Oncology Department, Ramón y Cajal University Hospital, IRYCIS, Alcalá University, 28034 Madrid, Spain; 4Centro de Investigación Biomédica en Red de Cáncer (CIBERONC), 28029 Madrid, Spain

**Keywords:** colorectal cancer, tumor microenvironment, diet

## Abstract

Colorectal cancer (CRC) is one of the most common cancers in Western countries and remains the second most common cause of cancer death worldwide. Many studies show the importance of diet and lifestyle in the incidence of CRC, as well as in CRC prevention. However, this review summarizes those studies that analyze the impact of nutrition on tumor microenvironment modulation and cancer progression. We review the available information about the effects of specific nutrients on cancer cell progression and on the different cells within the tumor microenvironment. Diet and nutritional status in the clinical management of colorectal cancer patients are also analyzed. Finally, future perspectives and challenges are discussed, with a view to improving CRC treatments by employing nutritional approaches. These promise great benefits and will eventually improve CRC patients’ survival.

## 1. Introduction

Although death rates from colorectal cancer (CRC) have declined in recent years due to treatment efficacy, CRC remains the second most common cause of cancer death worldwide [1,2]. CRC accounts for almost 10% of cancer deaths, second only to lung cancer, with approximately 18% of deaths.

Moreover, the incidence of CRC has increased greatly in the last decade, most rapidly in recent years and among the younger population [3,4]. More and more new cases of CRC are being diagnosed, especially in highly developed countries, suggesting that there are external factors contributing to the early onset of CRC [5]. Although there are studies that indicate that the factors contributing to the early onset of CRC are not well known, many other studies indicate that lifestyle changes and type of food or diet are among the factors involved in the worldwide increase in CRC prevalence in recent years [2,6,7,8]. These changes are especially pronounced among the young population, which means there will be a great increase in CRC in young people in the near future [9].

There are many epidemiological studies indicating that factors related to changes in diet, lifestyle, obesity, and environment play a major role in the prevention of CRC [10,11,12,13]. Thus, it has been observed that in countries with a good diet and lifestyle habits, such as regular physical activity, the occurrence of colorectal cancer is 20–25% lower [14]. Some studies have shown that high consumption of simple sugars during adolescence results in an increased risk of rectal adenoma [15]. However, adherence to the Mediterranean diet, which is characterized by a low intake of sugar-sweetened beverages and red meat, as well as a high intake of fish, is associated with lower odds of developing advanced polyps. Likewise, diets rich in whole grains, fiber, fruits, and vegetables are associated with a lower risk of cancer [16]. However, meta-analyses do not fully support the association of a lower risk of CRC with fruit and vegetable consumption [17]. In contrast, it is widely accepted that the high consumption of meat and ultra-processed foods is associated with an increased risk of colorectal cancer [18,19,20]. In a similar way, a high intake of ultra-processed foods has been recently associated with colorectal adenomas, especially advanced and proximal adenomas [21]. Similarly, a higher prevalence of colorectal polyps is associated with the replacement of fresh food by canned food consumption [22].

During tumor progression, the intratumoral heterogeneity rises along with the maturation of the cellular and non-cellular components of the tumor microenvironment (TME). The TME is composed of extracellular matrix (ECM), stromal cells (such as fibroblasts, mesenchymal stromal cells, pericytes, blood, and lymphatic vascular networks), and immune cells (including T and B lymphocytes, natural killer cells and tumor-associated macrophages) [23]. These supporting cells, recruited from the local host stroma, promote extracellular matrix remodeling, cell migration, neoangiogenesis, invasion, drug resistance, and evasion of immunosurveillance through the production of various growth factors, chemokines, and cytokines [24]. Under favorable conditions, tumor cells are able to induce a switch in these stromal cells, turning them into tumor-associated stromal cells, which can secrete many pro-tumorigenic factors, including interleukin 6 (IL-6), IL-8, stromal-derived factor-1 alfa (α-SDF1), vascular endothelial growth factor (VEGF), tenascin-C and matrix metalloproteinases, among other factors, all of which recruit additional tumor and pro-tumorigenic cells to the developing microenvironment [25]. Thus, several interactions of cancer cells with their environment contribute to tumor progression and metastasis by enhancing vascularization, repressing the immune response, and inducing inflammation [26,27].

In summary, the epidemiological data of CRC patients provides a lot of information about diet and lifestyle. Similarly, a lot of information shows the association of diet styles with the incidence and prevention of CRC. However, there is not so much information about the effect of diet on tumor progression or about the clinical management of CRC patients. This article reviews this information, focusing on the effects of diet on the modification of the tumor microenvironment, which is needed for tumor progression.

## 2. Effect of Diet on the Development of Colorectal Cancer

### 2.1. Tumor Epithelial Cells

During tumor progression, cancer cells have to overcome several challenges, such as growth in the nutrient-altered and oxygen-deficient microenvironment of the primary site, intravasation into vessels where anchorage-independent growth is required, and invasion of distant organs where the environment changes during metastasis processes. Thus, cancer cells need metabolic reprogramming at every step of cancer progression [28,29].

Diet may influence cancer development through epigenetic mechanisms that could affect the expression of those genes involved in cell proliferation and growth [30,31] (Figure 1). Nutrient availability determines the abundance of certain energy metabolites that are essential co-factors for epigenetic enzymes. During CRC, aberrant epigenetic marks are accumulated, and epimutations are selected to drive tumorigenesis by causing transcriptome profiles that may diverge from the cell of origin [32]. In the DNA methylation pathway, the most straightforward is vitamin C, which is a cofactor for the ten-eleven translocation family of enzymes that mediate the formation of hydroxymethylation and eventual DNA demethylation [30]. An in vitro study showed that cultured CRC cells with KRAS or BRAF mutations are selectively killed when exposed to high levels of vitamin C. The increased uptake of the oxidized form of vitamin C, dehydroascorbate (DHA), via the GLUT1 glucose transporter [33] causes oxidative stress as intracellular DHA is reduced to vitamin C-depleting glutathione. Thus, ROS accumulates and inactivates glyceraldehyde 3-phosphate dehydrogenase (GAPDH). This GAPDH inhibition in highly glycolytic KRAS or BRAF mutant cells leads to energy depletion and cell death not seen in wild-type cells. In fact, in other types of cancer, such as thyroid cancer, it has already been demonstrated that vitamin C kills thyroid cancer cells by inhibiting MAPK/ERK and PI3K/AKT pathways via a ROS-dependent mechanism, suggesting that pharmaceutical concentration of vitamin C has potential clinical use in thyroid cancer therapy [34].

The ketogenic diet (KD) was first created in the 1920s as a treatment for uncontrollable epilepsy [45], consisting of a high-fat, low-carbohydrate diet with appropriate proteins and calories. The conventional KD is a 4:1 formulation of fat content to carbohydrate plus protein. 90% of the calories in a common 4:1 KD come 90% from fat, 8% from protein, and only 2% from carbohydrates. KD provides a fat-rich, low-carbohydrate diet during chemotherapy treatment that may decrease blood glucose levels and cause ketosis, starving cancer cells of energy while normal cells adjust their metabolism to use ketone bodies and survive. Thus, this glucose reduction may lead to a drop in insulin and insulin-like growth factor levels, which are involved in the proliferation of cancer cells. It is believed that cancer cell mitochondria have inefficient mitochondrial electron transport chain activities, which results in higher steady-state levels of O_2_/H_2_O_2_, as fatty acid oxidation occurs largely in the mitochondria. Additionally, they accelerate glucose metabolism to provide reducing equivalents that will offset the excess H_2_O_2_. According to the theory, restricting glucose intake and increasing reliance on fatty acid oxidation while consuming a KD would preferentially increase tumor cell sensitivity to radiation and chemotherapy over normal cell sensitivity through a process involving oxidative stress [26,31]. In an in vivo approach, a ketogenic formula of KD decreased tumor size in tumor-bearing mice [41]. Mice treated with the ketogenic formula showed higher blood β-hydroxybutyrate (βHB) levels, leading to an inhibition of plasma IL-6 concentration and, thus, an anti-inflammatory effect. Moreover, this study found that ketogenic formulae might suppress cancer progression and the accompanying systemic inflammation without adverse effects on weight gain or muscle mass, which might help to prevent cancer cachexia. Another study in BALB/c nude mice injected with the HCT116 cancer cell line showed that a KD rich in omega-3 fatty acids or in medium-chain triglycerides delayed tumor growth when compared with a standard diet group [42]. In fact, in a clinical trial with rectal cancer patients during radio-chemotherapy, KD caused an improvement in several biomarkers of metabolic health (gamma-glutamyl-transpeptidase, triglyceride-glucose index, HDL cholesterol/triglyceride ratio, and free triiodothyronine) in comparison with a standard diet group, with amelioration in the role, emotional and social functioning and, thus, in quality of life [44].

It is known that adherence to Mediterranean Diet (MD) eating habits correlate to decreased overall cancer mortality and lower cancer risk, especially CRC [35,46,47]. Olive oil polyphenols, red wine resveratrol, and tomato lycopene are secondary antioxidants produced by plants and widely consumed in the MD that show in vitro several ways of interfering with molecular cancer pathways [35]. One in vitro study showed that a phenol mixture extracted from virgin olive oil is able to inhibit initiation, promotion, and metastasis in CRC cell lines. The phenol mixture reduced the genotoxicity induced by H_2_O_2_ in HT29 cells, increased barrier function in CACO2 cells, and decreased the invasiveness of HT115 cells [36]. Resveratrol, a phytoalexin found in many plants, such as peanuts, berries, and grapes (and, thus, in red wine), may suppress the invasion and metastasis of colon cancer through inhibition of epithelial-mesenchymal transition via the protein kinase B (AKT)/glycogen synthase kinase 3β (GSK3β)/Snail signaling pathway [37]. Thus, there is an increase in E cadherin expression and a decrease of N cadherin, phospho (p)-AKT1, p-GSK3β, and Snail in colon cancer. Lycopene (a carotenoid found in tomatoes, other vegetables, and fruits), sulforaphane (an isothiocyanate present in cruciferous plants), quercetin (found in a wide variety of vegetables and fruits), and curcumin (a turmeric polyphenol) are other phytochemicals abundantly found in the MD. When applied together, they are able to inhibit proliferation and decrease the metabolic activity of colon cancer cell lines in vitro through inhibition of DNA synthesis while simultaneously not being toxic to normal colon epithelial cells [38]. Moreover, this phytochemical mix enhances the antiproliferative activity of 5-fluorouracil (5-FU) and cisplatin, which suggests the chemopreventive potential of this proposed combination of natural substances and the value of these compounds as adjuvants during chemotherapy treatment. In fact, several natural compounds, currently used all round the world but traditionally considered Chinese medicines, significantly improve life quality and combat cancer by creating synergies in CRC chemotherapy administration, enhancing the chemopreventive functions of 5-FU and reducing its side-effects. These products include quercetin, ginseng (with ginsenosides, flavonoids, polypeptides, and polysaccharides), and tea (derived from the leaves of Camellia sinensis, rich in catechins and polyphenols) [39]. Numerous studies have shown that Tea polyphenols (TPs) can alter a number of signaling pathways in cancer cells, including the mitogen-activated protein kinase pathway, PI3K/Akt pathway, Wnt/β-catenin pathway, and 67 kDa laminin receptor pathway, to inhibit proliferation and promote cell apoptosis [48]. Moreover, diet-derived phytochemicals, such as polyphenols and flavonoids, can modulate both gut microbiome and colon CSCs, with a decrease in their tumorigenic effects. These molecules, present in countless vegetables, are able to increase many colonic bacteria (Firmicutes, Bacteroidetes, Actinobacteria, and Proteobacteria) and metabolize them to produce several by-products that alter the pH of the colon environment and maintain the balance of the colonic microbiome, which may contribute to host well-being. These phytochemicals and their by-products can also interfere in the preservation of CSCs or modify the CSC through various regulatory signaling pathways such as Notch, Hedgehog, and Wnt/β-catenin [40].

A lifestyle consistent with the American Cancer Society (ACS) guidelines, which include maintaining healthy body weight, engaging in physical activity, and eating a diet rich in vegetables, fruit, and whole grains, was linked to a 42% lower risk of passing away during the study period, according to a cohort study of 992 patients with colon cancer [43]. Patients with high adherence to the recommendations had an 85% 5-year survival probability, while those with low adherence had a 76% 5-year survival probability and a 9% absolute reduction in 5-year mortality risk.

### 2.2. Stromal Cells

#### 2.2.1. Cancer-Associated Fibroblasts (CAFs) and Mesenchymal Stem Cells

Fibroblasts are the cells responsible for producing the extracellular matrix of connective tissue, so they are essential for preserving the structural integrity of most tissues [49]. Within the tumor microenvironment, the most abundant type of cells are fibroblasts, which under these conditions are called cancer-associated fibroblasts (CAFs). CAFs present a myofibroblast phenotype, expressing markers such as alfa-smooth muscle actin (α-SMA), fibroblast-specific protein 1 (FSP1), fibroblast-activating protein (FAP), and platelet-derived growth factor receptor alfa and beta (PDGFRα and β) [50,51,52]. In the tumor microenvironment, activated non-malignant cells, including CAFs, exhibit a secretory profile under stress conditions, which may lead to cancer progression and chemoresistance [53], stimulating hyperproliferation and progression of preneoplastic epithelial cells and accelerating tumorigenesis by neoplastic epithelial cells [54]. Some studies have shown the effects of different diet patterns on these stromal cells (Figure 2A).

The MD promotes a high diet of fruits, vegetables, whole grains, legumes, olive oil, and fish, a low intake of saturated fats such as butter and other animal fats, red meat, poultry, and dairy products, as well as a regular but moderate red wine consumption at meals [69]. Several natural substances have rejuvenating and anti-oxidant capabilities in this context. For instance, quercetin, the most prevalent and abundant natural flavonoid, is found in many different fruits and vegetables, including shallots, celery, lettuce, tomatoes, capers, red onions, and cruciferous vegetables. Refs. [55,59], rejuvenate senescent fibroblasts by resetting efficient proteolysis through the activation of 26S proteasome functions via the Nrf2 transcription factor, which activates the transcription of a set of antioxidant and anti-inflammatory genes [55]. Carotenoids are colorful liposoluble pigments with antioxidant properties that are present in many foods, for example, fruit, vegetables, and fish [70]. In vitro, studies have demonstrated that β-carotene (BC) treatment suppressed expression of the α-SMA and vimentin fibroblast activation markers in a normal colonic fibroblast cell line (CCD-18Co) [56]. Conditioned media from BC-treated activated fibroblasts also inhibited colon cancer cell invasiveness, migration, and the epithelial-mesenchymal transition (EMT) in a human colon cancer cell line (HCT116) through the downregulation of transforming growth factor β1 (TGF-β1) and α-SMA, and the upregulation of E-cadherin.

In contrast, a study using an in vivo mouse model and ex vivo organoid experiments showed that mice fed a high-fat diet (HFD) had deregulated colonic stem cells and mesenchymal stem cells (MSCs) [57]. Colonic MSCs dysregulation due to HFD eating included an increase in cell count, Wnt2b overexpression, and the activation of CAF-like characteristics. Colonic MSCs produced from HFD consistently induced Wnt-related signals and cancer stem cell hallmarks.

#### 2.2.2. Endothelial Cells

Endothelial cells are the main structural component of blood vessels. During tumor progression, endothelial cells undergo an angiogenic switch, which has long been considered to be dictated by angiogenic activators, such as VEGFs, IL-8, tumor necrosis factor α (TNF-α), and hypoxia-inducible factors (HIFs) [26] and other signaling pathways (e.g., Notch). However, recent findings show that a metabolic switch in the endothelial cells also drives angiogenesis [71].

In terms of nutrition (Figure 2B), the anti-tumor effect of the ketone bodies acetoacetate and βHB has been shown in several colon and breast cancer lines in vitro through glycolysis inhibition and the over-expression of uncoupling protein 2 (UCP2), which also inhibits mitochondrial ATP generation, leading to a consequent reduction of ATP concentration [72]. Moreover, ketone bodies affect stromal cell metabolism, as in endothelial cells. In fact, it has been demonstrated that, in mouse glioma models, a KD or caloric restriction reduced tumor microvasculature with a significant reduction of HIF-1α and VEGF receptor two levels [58].

Furthermore, olive oil and red wine polyphenols contained in the MD reduce inflammatory angiogenesis, a key pathogenic process in cancer and atherosclerosis, in human cultured endothelial cells through the inhibition of COX-2 protein expression, prostaglandin production, and MMP-9 release. This effect is accompanied by a substantial reduction in reactive oxygen species (ROS) levels and NF-κB activation [73]. For example, oral consumption of oleuropein, the main phenolic compound in olive oil, results in fewer blood vessels providing strong anti-angiogenic properties [60]. Ellagic acid is a phenolic substance that is contained in many red fruits, including pomegranate peel, raspberries, strawberries, and cranberries, as well as in walnuts and goji berries. Thus, ellagic acid has an anti-angiogenic effect mediated by the abolition of the hypoxia gate at the phosphatidylinositol-3-kinase/AKT/mTOR, MAPK, and VEGF/VEGFR2 pathways, which trigger the suppression of the response towards HDAC6 and hypoxia-inducible factor 1 subunit alfa (HIF1α) [59].

#### 2.2.3. Immune Cells

The immune system, which is made up of innate immune cells such as neutrophils, macrophages, dendritic cells (DCs), mast cells, and natural killer (NK) cells, as well as adaptive immune cells such as T- and B-lymphocytes, plays a role in both preventing and promoting the growth of tumors by acting in both a pro- and anti-tumor manner [74]. Similarly to this, various diet components may have an impact on how CRC patients’ immune systems are monitored (Figure 2C).

Several in vitro and in vivo studies provide evidence that the KD and ketone bodies (especially βHB) are able to suppress the NLRP3 inflammasome and to reduce levels of inflammatory factors such as TNF-α, IL-1, -6 and -18 and prostaglandin E2, with the consequent anti-inflammatory effect [26]. A KD inhibited the rise in plasma IL-6 and subsequent progression of inflammation in a colon cancer mouse model [41]. As a result, the KD appears to produce a metabolic environment that is adverse to the development of cancer cells [26]. Additionally, research using mice models of pancreatic and glioma cancer has demonstrated that the KD enhances the immune response against cancer progression [75,76].

TPs can prevent the growth and spread of CRC by their anti-inflammatory, anti-oxidative or pro-oxidative, and pro-apoptotic effects, which are accomplished by modulations at multiple levels. Furthermore, according to recent research, TPs may inhibit the growth and metastasis of CRC by modifying the diversity of the gut microbiota to strengthen the immune system and reduce inflammatory responses, as well as by encouraging the proliferation of T lymphocytes and reducing M2 macrophages [48]. Through the downregulation of CD163, Arg1, TGF-1, and PPAR-g M2-associated markers in a human monocytic leukemia cell line (U937 cells), in vitro methods have demonstrated that BC suppresses M2 macrophage polarization [56]. Working in the same line, in vivo studies showed that BC supplementation inhibited the expression of the CD206, Arg1, Ym-1, CD163, and PPAR-g M2 macrophage markers in an azoxymethane/dextran sodium sulfate (DSS)-induced colitis-associated CRC mouse model. It has been demonstrated that black raspberry components and their benzoate metabolites, which are also created during intestinal metabolism of many commonly consumed plant-based foods, such as fruit and vegetables and whole grains, upregulate Smad4 to suppress preneoplastic colonic epithelium and improve NK cell function to slow the progression of CRC [61].

Iron is a vital micronutrient present in meat, fish, legumes such as beans and lentils, and vegetables such as spinach [77] that is required for cell proliferation since DNA synthesis requires the iron-dependent enzyme ribonucleotide reductase. Excessive intestinal iron within the gastrointestinal tract is a risk factor for colorectal cancer development through its favoring cancer cell proliferation [63]. It has been suggested that vitamin C, whose sources are mainly fruit such as citrus fruit, kiwi, and mango, and vegetables such as broccoli, tomatoes, and peppers, has a protective role against cancer by chelating iron, leading to greatly enhanced absorption of iron from the diet [62]. However, iron-deficient anemia (IDA), which is common in patients with colorectal cancer, may alter immune-cell function, leading to a decline in the immune system’s immunosurveillance, which, in turn, could aid tumor development [63]. In addition, iron deficiency may result in iron-limited erythropoiesis [64] and may also alter immune cells in the tumor microenvironment, causing them to exert a pro-tumorigenic response [63]. The effects of the iron shortage on anti-inflammatory CD4+ Regulatory T cells have been examined in vitro, and it was discovered that iron chelation led to defective Treg activation and proliferation [65], which may promote the loss of the immunosuppressive effects of these cells, thus leading to the chronic inflammation associated with colorectal cancer. Moreover, other in vitro studies have shown that M2 macrophages had an expression profile (ferroportin upregulation and the downregulation of H ferritin and heme oxygenase) that enhanced iron release, whereas M1 macrophages showed changes in gene expression (ferroportin repression and H ferritin induction) that favor iron sequestration [66]. Tumor-associated macrophages (TAM), whose phenotype closely resembles that of M2 macrophages, favor tumor growth by an immunosuppressive and tumor-promoting phenotype characterized by a distinct repertoire of growth factors, cytokines, and chemokines. In terms of therapy, oral iron therapy is the current standard treatment for IDA, but this treatment increases the concentration of luminal iron in the gastrointestinal tract, which increases oxidative stress and inflammation that may contribute to the progression of colorectal cancer. In contrast, intravenous iron therapy in CRC patients with IDA has shown a replenishment of iron stores without a rise in intestinal iron concentration [77].

On the other hand, it has been shown that HFD-induced obesity affects CD8+ T cell activity in the TME, increasing tumor growth in a CRC mouse model [67]. Adipose tissue is overpopulated with a range of pro-inflammatory immune cells when it is in a hyperplastic and hypertrophic condition, including classically activated macrophages, NK cells, mast cells, neutrophils, DCs, B cells, CTL, and T helper 1 (Th1) cells. These cells discharge pro-inflammatory substances such as tumor necrosis factor-α (TNF-α), interferon-γ (IFN-γ), IL-1β, and IL-6, which exacerbate local and systemic inflammation and act as potent tumor promoters [68].

### 2.3. Gut Microbiota

Human gut microbiota has been extensively researched in recent years, and, especially since the advent of metagenomic investigations, our understanding of the resident species and their potential applications has grown. The human gut contains trillions of microorganisms, including bacteria, archaea, fungi, protists, and viruses, with bacteria serving as the majority of inhabitants. The human gut is one of the body’s most intricate networks. Consequently, the phyla Proteobacteria, Firmicutes, and Bacteroidetes are typically found in the colon [78,79]. The crosstalk between gut microbiota and its metabolites with the TME affects the host immune system and intestinal epithelium and promotes or inhibits the development of tumors [80].

The involvement of probiotics and prebiotics in the maintenance and status of gut microbiota is known; some studies have determined their involvement in CRC progression (Figure 3). Prebiotics are non-digestible food components that show beneficial effects on the host by stimulating the growth and/or activity of probiotics in the colon after fermentation. These components are mainly dietary fibers, which resist hydrolysis by human digestive enzymes. They include celluloses, hemicelluloses, pectins, gums and mucilages, inulines, and oligosaccharides. They are present in a wide variety of fruit and vegetables, and their fermentative products include short-chain fatty acids (SCFAs) [81]. In vivo, studies have revealed the beneficial effects of combination therapy of galacto-oligosaccharides and inulin prebiotics against the development of CRC in a 1,2-dimethyl hydrazine dihydrochloride (DMH)-induced rodent model, showing the inhibition of aberrant crypt foci formation and higher levels of SCFAs [82]. Prebiotics have also demonstrated anti-tumor properties by down-regulating the expression of COX2, iNOS, NF-kB, and gastrointestinal glutathione peroxidase levels due to their biogenic effects and immunomodulatory role. They modulate microbiota by inhibiting pathogenic multiplication and enhancing cell apoptosis.

Probiotics synergizing with prebiotics have a therapeutic impact on CRC [87]. Probiotics are live microorganisms that have beneficial effects on the health of the host. Widely used probiotics are yeast strains of Saccharomyces cerevisiae and bacterial species of Lactobacillus, Bifidobacterium, Streptococcus, Lactococcus, and Escherichia coli [81]. Food-based probiotic products include dairy products (e.g., cheese, yogurts, ice cream, milk, acidified milk and creams), meats and meat products, bread or other fiber snacks, chocolate, fruit juice, and other fruit preparations [88]. For example, yogurts contain bacteria from Bifidobacterium sp., Lactobacillus sp., and Streptococcus sp., and kefir includes Bifidobacterium sp., Lactobacillus sp., Leuconostoc sp., and Saccharomyces sp. [89]. A study showed that a probiotic mixture treatment (consisting of Lactobacillus and Bifidobacterium species, resistant dextrin, isomalto-oligosaccharides, fructose oligosaccharides, and stachyose) decreased the proliferation, migration, and invasion of CT26 colon cancer cell line in vitro as well as in BALB/c mice in vivo approach [90]. Specific probiotic strains of Bifidobacterium infantis and Bifidobacterium breve activate intestinal DCs through toll-like receptors (TLRs) and induce retinoid acid metabolism, driving the expression of Foxp3+ Tregs and type-1 regulatory T cells (Tr1) and IL-10 release [85]. Some other probiotic bacteria, such as Lactobacillus rhamnosus GG and Lactobacillus acidophilus, downregulate the expression of Th17 cells and secretion of IL23 and IL17 via inhibition of STAT3 and NF-κB signaling or induce switch of macrophage phenotype from pro-inflammatory M1 to immunosuppressive M2. In fact, a clinical trial demonstrated that a probiotic administration (Lactobacillus, Bifidobacterium, and Streptococcus species) causes a statistically significant reduction in postoperative complications in the location of tumors in the rectum (−33.3%) and the ascending colon (−16.7%) [91]. There are also approaches that study the symbiotic combination of prebiotics and probiotics in the treatment of colorectal cancer. For example, an in vivo study revealed that the combination of Djulis (Chenopodium formosanum, a grain containing prebiotic dietary fiber) and Lactobacillus acidophilus had an enhanced effect on the Bax/Bcl2 ratio and Caspase 3 expression and reduced levels of inflammatory-related COX2 in a DMH/DSS-induced CRC rat model [86].

Many microbiota-derived metabolites, also called “postbiotics,” are expected to act as signaling molecules that affect biological processes and have a positive impact on the host, either directly or indirectly. These metabolites include substances such as bile acids, vitamins (K, B12), amino acids, serotonin, hypoxanthine, and SCFAs [85,92,93]. One study demonstrated that green TPs could induce the production of SCFAs by so-called “beneficial” bacterial genera, such as Faecalibacterium, Blautia, Bifidobacterium, Roseburia, Eubacterium, and Coprococcus, which reduce the functional markers of inflammation [48]. The microbial production of SCFAs in sufficient quantities appears to be a hallmark of health-promoting diets. A main source of SCFAs in the intestine is the anaerobic fermentation of dietary fiber, composed of polysaccharides that are indigestible to the host [94]. The main metabolites produced in the colon by bacterial fermentation of dietary fibers are acetate, propionate, and butyrate, which are structurally similar to the ketone body βHB [81,95,96,97]. Two main bacterial groups produce SCFAs. Thus, Bacteroidetes mainly form propionate and acetate, while Firmicutes produce butyrate [97]. Butyrate can be directly consumed within certain food (such as butter and cheese), but major sources comprise dietary fiber fermentation through butyrogenic bacteria (mostly by Firmicutes phylum), fermentation of acetate and lactate obtained by cross-feeding from other microbial species or degradation of mucin by certain bacteria, including the Clostridia species Rosburia intestinalis and Eubacterium rectale. Butyrate is the main energy source for healthy colonocytes, accounting for about 70% or more of their energy requirements through β-oxidation. It is preferred to ketone bodies, glucose, and glutamine [94,98]. Butyrate regulates gene expression and induces hyperacetylation of histones by the inhibition of histone deacetylase (HDAC) [83], which triggers apoptosis in tumor cells through the expression of tumor suppressor genes p21 and p53 [84]. In addition, the butyrate induces strong production of effector molecules in cytolytic T lymphocytes (CTLs) such as CD25, IFN-γ, and TNF-α, resulting in increased anti-tumor reactivity and improved therapeutic outcomes [99]. Fat primarily impacts the composition of the gut microbiota by increasing bile acid output, and the microbiome is necessary for bile acids to enter the colon and be processed into secondary bile acids, which may have carcinogenic potential when combined with colonic butyrate deficiency [100]. In this context, a study in rats reported that a high-fat diet significantly increased the population of Firmicutes and decreased the population of Bacteroidetes, whilst dietary polyphenols derived from aronia, haskap, and bilberry promoted a beneficial rise in the population of both Firmicutes and Bacteroidetes, with some approaching low-fat diet levels [101]. A study in mice revealed that changes in the microbiota composition caused by an HFD are characterized by a decrease in Lactobacillales and by an increase in the Clostridium subcluster XIVa, which were suppressed by agaro-oligosaccaride supplementation [102].

## 3. Nutritional Interventions during the Clinical Management of CRC Patients

Malnutrition at cancer diagnosis ranges from 15–40% and increases during treatment to 40–80% [103]. This condition affects all stages of cancer treatment [104], although patients with advanced tumors are at greater risk of malnutrition due to inflammation, calory depletion, and malabsorption [105].

Around 35% of patients with CRC present preoperative malnutrition [106], which is the best predictor of the risk of postoperative complications. During chemotherapy or radiotherapy, malnutrition increases the risk of toxicity, worsens the quality of life, and reduces functionality. In addition, obesity is a potential source of inflammation, and as obese patients with CRC can develop malnutrition, it is important to apply these recommendations in this population, too [107].

The efficacy of nutritional interventions is related to the initiation and timing of the intervention and requires individual assessment and personalization [108]. Various validated scales such as MUST, NRS 2002 or MNA [109], anthropometric measurements, weight loss assessment [110], body composition assessment, muscle mass calculation [111,112], biochemical data, the calorimetric index [113], sarcopenia assessment [114] and quality of life should all be taken into account for the assessment. Recommendations during oncological treatment are to maintain weight and to eat a balanced diet that includes a variety of foods with a focus on protein, in addition to dietary supplementation and physical exercise (prehabilitation) [87] (Figure 4).

In order to prevent sarcopenia, the recommended nutritional support involves high protein content with glutamine, leucine, and omega-3 fatty acids, in addition to orexigenic agents and hormonal drugs contributing to the preservation of muscle mass [115,116]. Although low testosterone levels are associated with a high prevalence of sarcopenia, supplementation has not been shown to be effective in elderly patients with sarcopenia [117]. Ghrelin and ghrelin analogs stimulate appetite and muscle anabolism [117] and, together with angiotensin-converting enzyme inhibitors, IGF1 and myostatin inhibitors, are under study [118]. Periodic re-evaluation is required during the intervention.

### 3.1. CRC in Surgery

A systematic review published in 2017 analyzed nine studies with more than 500 patients [119]. Five of these compared prehabilitation and standard management in CRC. In relation to nutritional intervention, only two studies recommended evaluation by a nutritionist and the establishment of a diet with a daily protein amount of 1.2 g/kg of weight administered within one hour of physical exercise [120,121]. Another study analyzed prehabilitation treatment in 100 patients one month before surgery by administering 1.2 g/d of omega-3 fatty acids rather than 0.7 g/d, with this increase decreasing postoperative complications [106,122]. In addition, patients who received arginine, omega-3 fatty acids, and ribonucleic acid supplements experienced better immune response, oxygenation, and microperfusion than patients with conventional formulae [123]. In another randomized study with 60 patients with nasogastric tube nutrition (Impact), their nutritional and immunological status was better than that of those who received standard supplementation, which reduced their hospital stay [124]. The Taiwan Society of Surgeons developed a consensus on anti-inflammatory nutritional intervention in patients with CRC. They analyzed nine studies in which EPA consumption ranged from 1.2 g/d to 2.2 g/d and showed weight gain or stabilization [125]. They recommended taking at least 2.2 g/d of EPA. Consensus recommendations for patients at high risk of malnutrition prior to treatment include oral or parenteral nutrition of 25–30 kcal/kg/d and 1.2–1.5 g protein/kg/d to maintain or restore weight (GRADE: high). Oral nutrition containing n-3 PUFAs or lipid emulsion in parenteral nutrition should be added for patients with NLR ≥ 3 (GRADE: very low). After treatment, isocaloric oral or parenteral nutrition 25–30 kcal/kg/d and isonitrogenous 1.2–1.5 g protein/kg/d are recommended (GRADE: high), and n-3 PUFAs should be used with NLR ≥ 3 (GRADE: very low). Leucine, isoleucine, and valine balance the nitrogen imbalance produced by stress. These supplements increase superoxide dismutase, a key in the antioxidant pathway [126].

### 3.2. CRC in Oncological Treatments

Some studies have demonstrated the role of probiotics in patients receiving radiotherapy and chemotherapy [127], such as supplementation with Lactobacillus rhamnous decreasing diarrhea and abdominal pain [128]. A clinical trial on the efficacy and tolerability of Lactobacillus rhamnous in patients with radio-induced diarrhea showed that patients who had received probiotics had better fecal consistency and decreased bowel movements. In chemotherapy treatment, side effects such as dysgeusia, nausea, vomiting, anorexia, mucositis, dysphagia, and diarrhea are the major causes of poor tolerance to treatment and cause malnutrition. Targeted therapies and immunotherapy produce multiple side-effects such as asthenia, anorexia, mucositis, diarrhea, dysgeusia, or dysphagia. However, the full effect of nutritional status in patients with these treatments and possible nutritional interventions has not been determined [105].

## 4. Conclusions and Future Challenges

CRC remains the second leading cause of cancer death worldwide. In recent decades, a strong association between diet and CRC incidence has been studied and found. Many studies show that the Mediterranean diet reduces CRC incidence. Moreover, some studies also determined the reduction of tumor recurrence and protection from secondary tumors under specific diet patterns [129]. Otherwise, the association between CRC and the microbiome is widely known. Although many studies point to a complex interrelation between diet, gut microbiota, and CRC, most of these studies focused on diet as a preventive tool for CRC. Moreover, due to the long latency of colorectal carcinogenesis, it is difficult to discover with any profundity the specific effects of diet patterns on CRC progression and how these patterns could improve current therapeutic strategies. However, although there is not so much information about the effect of diet on tumor progression (most studies focus on its effect on the tumor microenvironment), it is clear that dietary management is essential in the clinical management of CRC patients. The economic impact of these nutritional interventions on gastrointestinal tumors has been evaluated. One study analyzed eight studies, two of them specific to CRC cancer, that demonstrated the savings of diet intervention. Two of the studies had a strong level of evidence. Nutritional interventions include oral dietary modifications, enteral nutrition, and parenteral nutrition, especially in the perioperative setting. Perioperative immunonutrition and recovery strategies after surgery have significantly decreased postoperative complications and decreased hospital stays. Savings due to the adoption of these measures have been estimated at more than 200 million dollars [130].

More clinical, preclinical, and basic research is required to determine the most beneficial food patterns to be administered during CRC in order to improve anticancer treatments and improve patients’ survival. These further studies should investigate the molecular mechanism of diet components in the tumor microenvironment and, thus, in tumor progression. They should also combine standardization of the microbiome studies of high-dimensional omics and epidemiological data. In short, our commitment to improving the survival and quality of life of CRC patients must be supported by the combination of the mentioned studies’ approaches and the development and establishment of early dietary clinical trials to support the translation of basic and preclinical research into clinical patients’ care. In the near future, the interaction between the diet, the microbiome, and target therapies to control the tumor microenvironment is likely to become powerful weapons in fighting CRC.

## Figures and Tables

**Figure 1 ijms-24-07317-f001:**
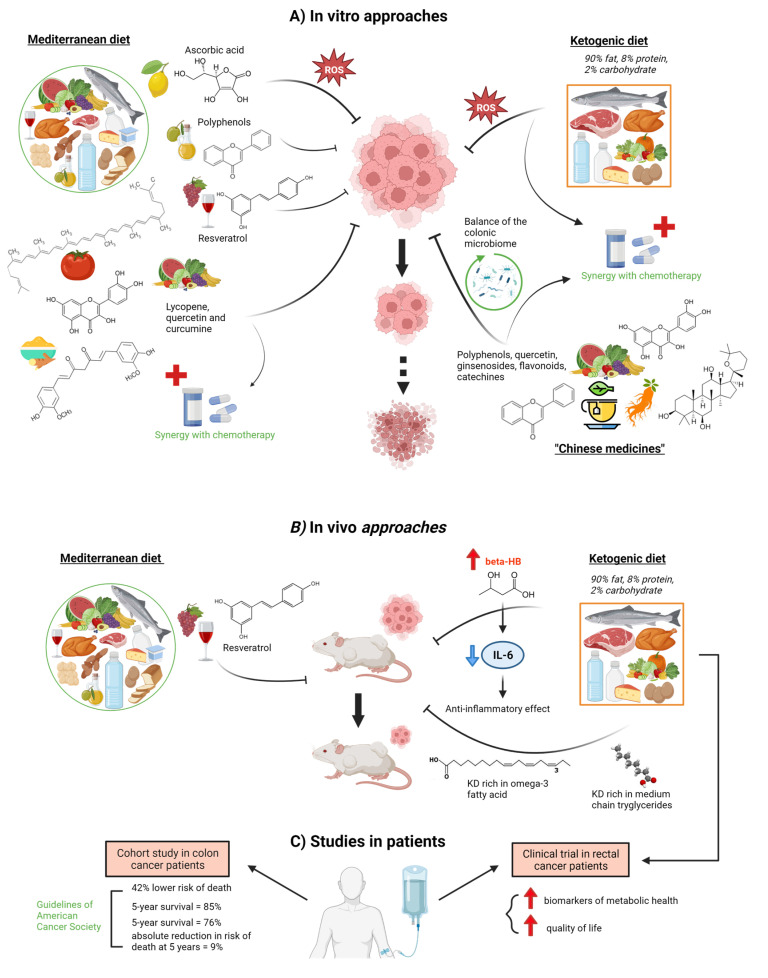
Effects of various diet components derived from the Mediterranean diet, Ketogenic diet, or Chinese medicine on colorectal cancer cells. (**A**) In vitro experiments showed the effects of Mediterranean diet components such as ascorbic acid, polyphenols, resveratrol, lycopene, quercetin, and curcumin on the proliferation of CRC cells [33,35,36,37,38]. Moreover, lycopene, quercetin, and curcumin also showed synergy with chemotherapy treatments [38]. Similarly, a ketogenic diet enhances tumor cell vs. normal cell sensitivity to radiation and chemotherapy by a mechanism involving oxidative stress [26,31]. In addition, Chinese medicines are able to improve life quality and exert anti-cancer effects by exerting synergies in CRC chemotherapy administration as well as modulating the microbiome balance, which contributes to host well-being [38,39,40]. (**B**) Through in vivo experiments, the inhibition of the invasion and metastasis of colon cancer by resveratrol [37] and the effects of a ketogenic diet on inflammation and cancer progression, and tumor inhibition have been demonstrated [41,42]. (**C**) Clinical studies showed that a lifestyle consistent with the American Cancer Society (ACS) guidelines is associated with better survival in colon cancer patients [43] as well as a better quality of life in rectal cancer patients on a ketogenic diet [44]. Dot arrow means possibility event. Created with BioRender.com (accessed on 14 April 2023).

**Figure 2 ijms-24-07317-f002:**
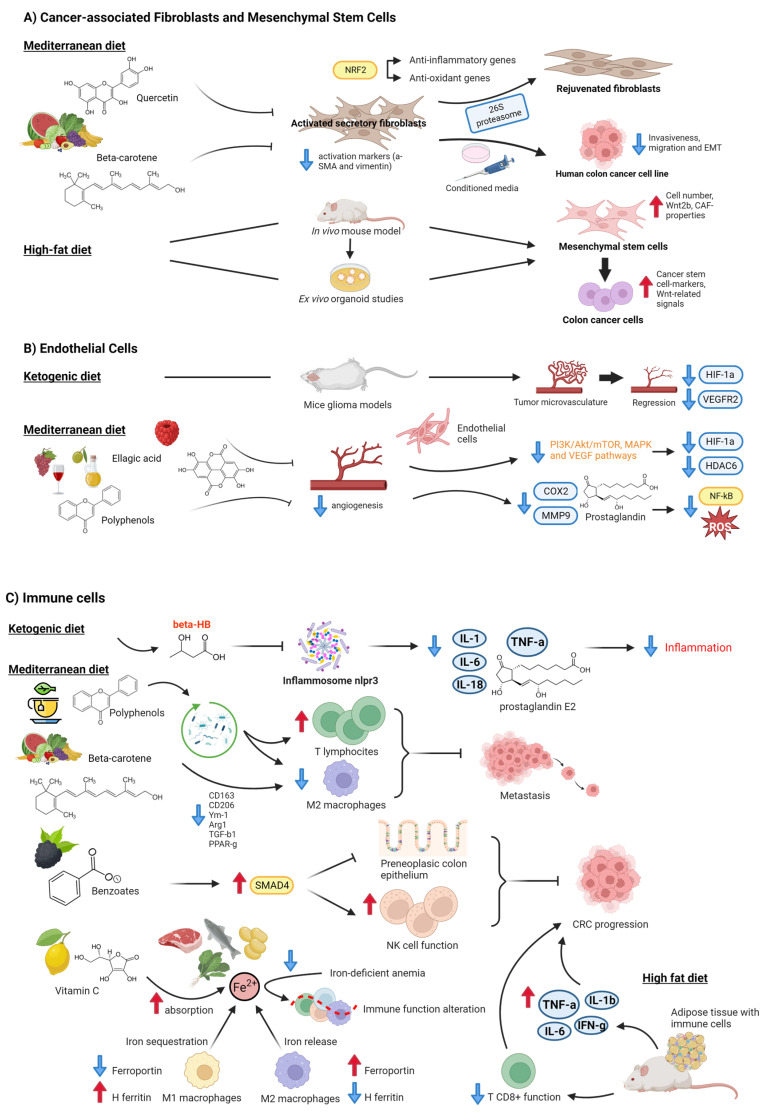
Effects of different diet patterns on stromal cells in colorectal cancer. (**A**) Cancer-associated Fibroblasts and Mesenchymal Stem Cells. Components of the Mediterranean Diet, such as quercetin or beta-carotene, rejuvenate senescent fibroblasts [55] and repress CAF-activated markers such as α-SMA and vimentin [56]. In contrast, a high-fat diet induces dysregulation of colonic stem cells and mesenchymal stem cells [57]. (**B**) Endothelial Cells. The ketogenic diet reduced the tumor microvasculature [58], as did ellagic acid [59], and polyphenols from the Mediterranean diet showed anti-angiogenic properties [59,60]. (**C**) Immune cells. A KD had an anti-inflammatory effect [26,41]. Polyphenols [48] and carotenes [56] also exert anti-inflammatory effects by modulating the composition of gut microbiota and thus regulating T-cells and M2 macrophages. Benzoates from black raspberries enhance NK cell function to delay CRC progression [61]. Iron balance in the intestinal tract also regulates immunosurveillance ability [62,63,64,65,66]. Finally, a high-fat diet impairs CD8+ T cells and induces pro-inflammatory factors, which enhances tumor growth [67,68]. Created with BioRender.com (accessed on 14 April 2023).

**Figure 3 ijms-24-07317-f003:**
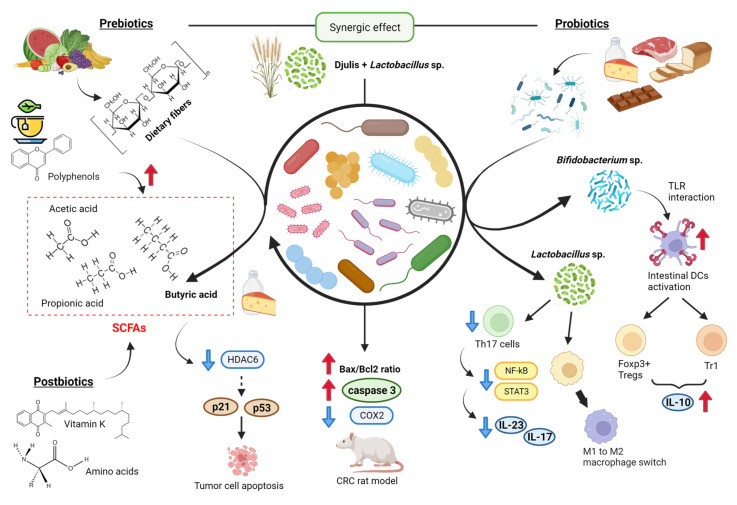
Prebiotics, probiotics, and postbiotics from the diet in CRC progression. Among prebiotics, some polyphenols and dietary fibers induce SCFA production by “beneficial” bacterial genera [48,81,83,84]. Some probiotics regulate the expression of Dendritic cells, Th17 cells, and the M1-M2 switch [85]. Finally, the combination of prebiotics and probiotics reduces levels of inflammatory-related COX2 in a model [86]. Created with BioRender.com (accessed on 14 April 2023).

**Figure 4 ijms-24-07317-f004:**
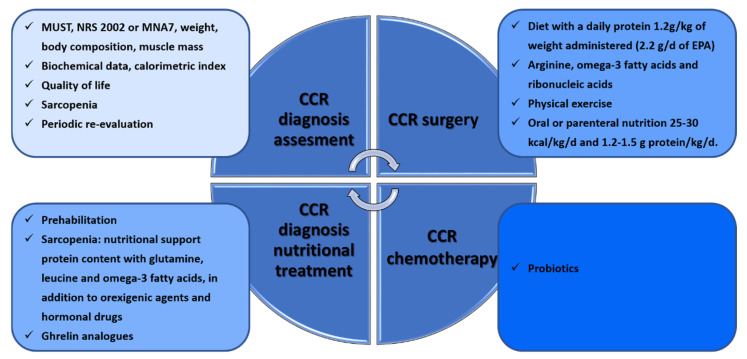
Workflow of nutritional interventions during CRC diagnosis and under surgery or chemotherapy therapies.

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
