# Peer review of "The Diet as a Modulator of Tumor Microenvironment in Colorectal Cancer Patients"

_ijms, 2023, doi:10.3390/ijms24087317_

Round 1
Reviewer 1 Report
This is a very well written in-depth review of the subject matter and will be a useful resource for readers.
The article would benefit from some minor edits.
1. Line 48 - "as high intake of fish, is associated with lower odds of advanced polyps" should be reworded to "as high intake of fish, is associated with lower odds of developing advanced polyps"
2. Line 396 - "of tumors on the rectum" should be reworded to "of tumors in the rectum"
3. Line 516 - " also determined the reduction of tumor recurrence and protection from second tumors" should be reworded as "also determined the reduction of tumor recurrence and protection from secondary tumors"
4. In the conclusion the section - "The economic impact of these nutritional in- 525 terventions in gastrointestinal tumors has been evaluated. One study analyzed 8 studies, 526 two of them specific to CRC cancer that demonstrated the savings of diet intervention. 527 Two of the studies had a strong level of evidence. Nutritional interventions include oral 528 dietary modifications, enteral nutrition and parenteral nutrition, especially in the periop- 529 erative setting. Perioperative immunonutrition and recovery strategies after surgery have 530 significantly decreased postoperative complications and decreased hospital stay. Savings 531 due to adoption of these measures have been estimated at more than 200 million dollars 532 [130]. The economic impact of these nutritional in- 525 terventions in gastrointestinal tumors has been evaluated. One study analyzed 8 studies, 526 two of them specific to CRC cancer that demonstrated the savings of diet intervention. 527 Two of the studies had a strong level of evidence. Nutritional interventions include oral 528 dietary modifications, enteral nutrition and parenteral nutrition, especially in the periop- 529 erative setting. Perioperative immunonutrition and recovery strategies after surgery have 530 significantly decreased postoperative complications and decreased hospital stay. Savings 531 due to adoption of these measures have been estimated at more than 200 million dollars 532 [130]." should be moved to the Introduction. This is new referenced information which should not be found in the Conclusion section of the manuscript.
5. Line 524 - "it is clear that management of diet is essential to clinical management of patients." should be reworded to "it is clear that dietary management is important in the clinical management of CRC."
6. Line 539 - The sentence "In short, our commitment to improving the survival and quality of life of CRC patients must be supported by the combination of these approaches and the development and establishment of early clinical trials to support the translation of basic and preclinical research into clinical patients’ care." suffers from no mentioning of dietary management here which is the whole premise of the manuscript. This is just a very generic statement provided here - early clinical trials could be any kind of trial such as chemotherapy or immunotherapy.
Author Response
In this review, Collado et al. provides a thorough evaluation of how nutrients affect the major constituents of the tumor microenvironment in colorectal cancer patients- including epithelial cells, stromal cells (fibroblasts), endothelial cells, immune cells and the microbiota. They discuss how clinical management of CRC patients have benefited from nutritional intervention. The autors conclude by presenting future challenges to better understanding the role of diet in colorectal carcinogenesis and leveraging nutrition supplementation to improve patient outcomes.
Overall, the review was well-written and the figures provided were informative and professionally presented. I only have minor comments that should be addressed and these are listed below
We appreciate the positive comments from the reviewer.
- Ensure that symbols (i.e. Alpha) are transmitted properly on the final manuscript. These were substituted with @ on the version that I reviewed.
We are grateful for this observation from the reviewer, which alerted us of this mistake. All symbols are now checked and correctly written.
- Include the appropriate reference immediately after the study was introduced. Below are some examples:
line 46, after “rectal adenoma”
line 95, after “glucose transporter”
line 120, after “intractable epilepsy”
line 135, after “tumor-bearing mice”
line 159, after “signaling pathway”
line 166, after “epithelial cells”
line 191, after “study period”
line 238, after “fibroblast cell line”
line 245, after “organoid studies”
line 300, after “leukemia cell line”
line 318, after “tumor development”
line 328, after “iron sequestration”
line 389, after “release”
line 475, after “500 patients”
line 484, after “hospital stay”
Following this suggestion the references places were changed.
- Reference 76 (line 72) is out of place. Need a better reference than #32 to support the statement on aberrant epigenetic marks and epimutations (line 91).
We are sorry but we do not understand the reviewer comment. There is not reference 76 in line 72. Could you please assess the correct numeration of references and lines?
- The sentence on probiotics (line 367) should be used to start the paragraph.
Following the reviewer comment, the sentence on line 367 has been used to start the paragraph.
Reviewer 2 Report
In this review, Collado et al. provides a thorough evaluation of how nutrients affect the major constituents of the tumor microenvironment in colorectal cancer patients- including epithelial cells, stromal cells (fibroblasts), endothelial cells, immune cells and the microbiota. They also discuss how clinical management of CRC patients have benefitted from nutritional intervention. The authors conclude by presenting future challenges to better understanding the role of diet in colorectal carcinogenesis and leveraging nutrition supplementation to improve patient outcomes.
Overall, the review was well-written and the figures provided were informative and professionally presented. I only have minor comments that should be addressed and these are listed below.
1. Ensure that symbols (ie. alpha) are transmitted properly on the final manuscript. These were substituted with @ on the version that I reviewed.
2. Include the appropriate reference immediately after the study was introduced. Below are some examples:
line 46, after “rectal adenoma”
line 95, after “glucose transporter”
line 120, after “intractable epilepsy”
line 135, after “tumor-bearing mice”
line 159, after “signaling pathway”
line 166, after “epithelial cells”
line 191, after “study period”
line 238, after “fibroblast cell line”
line 245, after “organoid studies”
line 300, after “leukemia cell line”
line 318, after “tumor development”
line 328, after “iron sequestration”
line 389, after “release”
line 475, after “500 patients”
line 484, after “hospital stay”
3. Reference 76 (line 72) is out of place. Need a better reference than #32 to support the statement on aberrant epigenetic marks and epimutations (line 91).
4. The sentence on priobiotics (line 367) should be used to start the paragraph.
Author Response
In this study, Collado et al. explain as the interaction between the diet, the microbiome and target therapies to control the tumor microenvironment, could improve the survival and quality of life of CRC patients.
Despite, many information about this approaches already exist, this is a well-written and thorough manuscript, useful to support the basic and preclinical research into clinical patient’s care.
We are happy to read the overall supportive attitude of the reviewer regarding the scientific content of our manuscript.
The autors should make minor corrections:
- To pay attention to abbreviations such as “Ketogenic Diet (KD)” (lane 121);
Thank you for this observation, all abbreviations have been checked though out the manuscript.
- There are unformatted symbols, as in lane 136, 187, 201 and others;
As in the previous point, thank you for alerting us about these mistakes. All symbols have been also checked.
- In lane 87 and 356, the paragraph to be justified and centered;
Both paragraphs have been justified. Sorry for this mistake.
- In lane 156 protein names are not indicated correctly; 158
We think that this is a mistake and the reviewer refers to proteins in lane 158. The name of the proteins are now correctly indicated.
- In lane 442 “rectaleamong”
This typographical mistake has been corrected. Thank you for the observation.
Reviewer 3 Report
In this study, Collado et al. explain as the interaction between the diet, the microbiome and target therapies to control the tumor microenvironment, could improve the survival and quality of life of CRC patients.
Despite, many information about these approaches already exist, this is a well-written and thorough manuscript, useful to support the basic and preclinical research into clinical patients’care.
The authors should make minor corrections:
- To pay attention to abbreviations such as “Ketogenic Diet (KD)” (lane 121);
- there are unformatted symbols, as in lane 136, 187, 201 and others;
- in lane 87 and 356, the paragraph to be justified and centered;
- in lane 156 protein names are not indicated correctly;
- in lane 442 “ rectaleamong “.
Author Response
This is a very well written in-depth review of the subject matter and will be a useful resource for readers.
We are thankful for the generous comments from this reviewer.
The article would benefit from some minor edits.
- Line 48 – “as high intake of fish, is associated with lower odds of advanced polyps” should be reworded to “as high intake of fish, is associated with lower odds of developing advanced polyps”
As suggested by the reviewer we have rephrased this sentence.
- Line 396 – “of tumors on the rectum” should be reworded to “of tumors in the rectum”
Thank you for alert us of this mistake. It has been corrected accordingly.
- Line 516 – “also determined the reduction of tumor recurrence and protection from second tumors” should be reworded as “also determined the reduction of tumor recurrence and protection from secondary tumors”
The wording correction has been realized following the reviewer suggestion.
- In the conclusion the section – “The economic impact of these nutritional interventions in gastrointestinal tumors has been evaluated. One study analyzed 8 studies, two of them specific to CRC that demonstrated the savings of diet intervention. Two of the studies had a strong level of evidence. Nutritional interventions include oral dietary modifications, enteral nutrition and parenteral nutrition, especially in the perioperative setting. Perioperative immunonutrition and recovery strategies after surgery have significantly decreased postoperative complications and decreased hospital stay. Savings due to adoption of these measures have been estimated at more than 200 million dollars [130].” Should be moved to the Introduction. This is new referenced information which should not be found in the Conclusion section of the manuscript”
We are sorry but we do not really agree with the aim of this comment. The economic impact of the diet is an issue not treated throughout the manuscript and it is just a commentary. Thus, we really think that the correct place is in the conclusion section as a future challenges. However, if the reviewer dos not agree we will be happy to move this paragraph.
- Line 524 – “it is clear that management of diet is essential to clinical management of the patients” should be reworded to “it is clear that dietary management is important in the clinical management of CRC.”
Following this suggestion, the sentence was rephrased.
- Line 539 – The sentence “In short, our commitment to improving the survival and quality of life of CRC patients must be supported by the combination of these approaches and the development and establishment of early clinical trials to support the translation of basic and preclinical research into clinical patient’s care.” Suffers from no mentioning of dietary management here which is the whole premise of the manuscript. This is just a very generic statement provided here – early clinical trials could be any kind of trial such as chemotherapy or immunotherapy.
This sentence was now rephrased following the reviewer recommendations.